# Do we need medical imaging-informed musculoskeletal models for simulations in healthy adults? A new workflow based on magnetic resonance imaging highlights the importance of personalized geometry

Ekaterina Stansfield[1], Willi Koller[2], Basílio Gonçalves[2], Hans Kainz[2]*

**1** University of Vienna, Faculty of Life Sciences, Department of Evolutionary Biology, Vienna, Austria,
**2** University of Vienna, Centre for Sport Science and University Sports, Department of Sport and Human Movement Science, Neuromechanics Research Group, Vienna, Austria

* hans.kainz@univie.ac.at

## Abstract

Musculoskeletal simulations often rely on generic models that may fail to accurately represent individual anatomy. While personalization using medical imaging can enhance model accuracy, it is often assumed to be more critical for pathological cases or pediatric populations, as generic models are typically based on healthy adults. However, even in healthy adults, generic models may not capture individual anatomical variability. In this study, we present a semi-automatic workflow for creating personalized musculoskeletal models based on magnetic resonance imaging (MRI). Our workflow concentrates on creating subject-specific joint centers and muscle paths. It also reconstructs bone surfaces without requiring MRI segmentation. It integrates 3D Slicer and Python scripts, and uses Thin-Plate Spline ('TPS') mapping of anatomically equivalent ('homologous') landmarks from generic models onto participants' anatomy. We applied this workflow to eight healthy participants, generating both generic-scaled and MRI-based models. Simulations were performed using participants' 3D motion capture data. Two model types were compared using a number of parameters, including model geometry, joint kinematics, dynamics, and resultant joint contact forces during one gait cycle. The results revealed clear geometric differences between the model types, with MRI-based models exhibiting a wider pelvis (mean distance between ischial bones was 98.0 ± 5.0 mm in generic-scaled and 11.0 ± 8.0 mm in MRI-based models) and distinct femur/tibia proportions (the mean ratio was 0.92 ± 0.040 in generic-scaled and 0.99 ± 0.033 in MRI-based models). MRI-based models captured systematic anatomical differences between males and females that were absent in generic-scaled models. These geometric differences substantially affected joint loading estimates. MRI-based models consistently produced higher peak joint contact forces with greater inter-individual variation, particularly at the knee joints. Early stance

**Data availability statement:** Code, models and motion capture data are available at https://github.com/katya-stanzy/thin-plate-spline_personalised_muscoloskeletal_model. Raw data required to replicate the results are available at https://doi.org/10.17605/OSF.IO/QANVW.

**Funding:** Project funded by the Austrian Science Fund (grant number P 35714-B). Ekaterina Stansfield received her salary from this grant. The funders had no role in study design, data collection and analysis, decision to publish, or preparation of the manuscript.

**Competing interests:** The authors have declared that no competing interests exist.

knee peak joint contact force was higher in the MRI-based compared with the generic-scaled model by $0.84 \pm 1.28$ body weights on average. Despite these differences in geometry and loading, joint kinematics were similar within individuals (mean difference was $0.8 \pm 2.47°$) and muscle moment arms aligned well with published cadaver data, supporting the validity of the personalization approach. This workflow simplifies the creation of MRI-based musculoskeletal models and challenges the assumption that personalization is unnecessary for healthy adults. The findings reveal significant sensitivity of joint contact forces to individual morphology, emphasizing the importance of personalized models even in healthy populations for biomechanical analyses.

## Author summary

Understanding healthy and pathological movement patterns in humans is essential for injury prevention, rehabilitation, and optimizing sports performance. Musculoskeletal modeling is a powerful tool for exploring these questions, as it allows researchers to study joint moments, muscle activation patterns, and joint contact forces. However, most studies rely on generic musculoskeletal models that are scaled linearly to match a participant's dimensions. This approach fails to accurately capture individual bone and muscle morphology. Personalization is often assumed to be more critical for pathological cases or children, while generic models are considered sufficient for healthy adults. To challenge this assumption, we developed a semi-automatic workflow for creating personalized musculoskeletal models based on magnetic resonance imaging (MRI). Our method avoids the need for bone segmentation and instead uses a non-affine fitting function to closely match individual geometry. By comparing MRI-based personalized models with generic-scaled models in eight healthy adults, we identified systematic biases in one of the most widely used musculoskeletal models. These findings demonstrate that even in healthy adults, personalization is important for accurately capturing individual anatomy and biomechanics. Our personalization pipeline is openly available, easy to implement, and designed to facilitate the use of highly personalized models in both clinical and research settings. This advancement has the potential to improve treatment planning, biomechanical assessments, and the overall accuracy of musculoskeletal modeling studies in the future.

## Introduction

Musculoskeletal simulations are widely used to investigate healthy and pathological movement patterns, providing insights critical for injury prevention, rehabilitation, and sports performance [1–3]. These simulations often rely on generic-scaled models, which are created by linearly scaling a pre-existing generic musculoskeletal model to match a participant's body dimensions. While this approach is convenient and widely

adopted, it assumes that generic models, which are typically based on healthy adults, are sufficient to represent individual anatomy, particularly in healthy populations. This assumption has led to the belief that personalization is primarily necessary for pathological cases or pediatric populations, where anatomical variability is more pronounced. However, even in healthy adults, individual differences in bone shape, muscle morphology, and joint geometry may significantly influence simulation outcomes.

Musculoskeletal simulations typically require motion data obtained from 3D gait laboratories, where reflective markers placed on the skin capture body segment locations in space. A personalized musculoskeletal model is then created by either modifying a generic model or building one from scratch [4–6]. These models, combined with motion data such as marker trajectories and ground reaction forces, enable simulations that estimate joint moments, muscle activation patterns, and joint contact forces [7,8]. However, generic-scaled models, which rely solely on skin marker positions to define scale factors, fail to account for individual anatomical differences, such as variations in bone shape, muscle-tendon paths, and joint surface geometry [9,10]. This limitation raises questions about the accuracy of generic models, even for healthy adults, and highlights the potential need for more personalized approaches.

Integrating medical imaging data into musculoskeletal models offers a pathway to improved accuracy by capturing individual anatomical features [11]. Personalization can range from simple adjustments, such as scaling bone lengths, to detailed reconstructions of muscle-tendon paths, muscle fiber properties, and joint surface features [6,12,13]. Achieving this level of personalization typically involves several steps, including determining joint center locations, aligning body segment orientations, and mapping muscle-tendon pathways to individual bone morphology. Approaches to muscle-tendon pathway personalization have included linear fitting of muscle via points [14], affine or non-affine mapping muscles onto bone surfaces [15], segmentation of muscle surfaces from 3D medical images [13], and population-based estimates followed by optimization algorithms [16,17]. Additionally, muscle fiber parameters, such as muscle fascicle length and pennation angle, can be personalized using 3D ultrasound and dynamic ultrasound tracking but these methods are restricted to superfical mucles [18,19]. While these methods improve anatomical accuracy, there is no single approach that is easily accessible to a student of musculoskeletal modelling.

Achieving accurate personalization requires methods that can capture individual anatomical variation beyond simple scaling. Most personalization approaches begin by creating 3D surface representations of participants' bones, which then serve as references for mapping muscle-tendon paths [17,20]. Small differences in muscle attachment sites were noted to affect muscle moment arms and, therefore, muscle force predictions to a great extent [21]. Non-linear (affine and non-affine) methods for this mapping have demonstrated superior accuracy compared to linear scaling approaches [17,22,23]. Despite the evidence of their superiority, non-affine personalization methods remain difficult to access and implement.

Several technical and practical barriers limit the widespread adoption of these more accurate approaches. First, most methods require bone segmentation from medical images to create the necessary 3D surface representations, which has been time-consuming until the recent development of AI segmentation tools. Second, sophisticated techniques such as Statistical Shape Analysis require access to large comparative populations of personalized models with matching muscle paths [20]. For example, the open-source optimization approach by Nolte et al. [17] employs population data from 34 individuals for non-affine mapping of femur and tibia muscle paths, but does not extend this capability to the pelvis. Third, while commercial software solutions, such as AnyBody [24] and Materialise [25], provide comprehensive non- affine mapping capabilities across all bone regions, equivalent open-source tools do not exist. Finally, most existing workflows require several days to create a personalized model, even for experienced researchers. These barriers prevent new users from moving beyond generic-scaled models, despite the known limitations of linear scaling approaches.

Recent advancements in open-source software, such as 3D Slicer [26] for annotation and OpenSim [27] for musculoskeletal modeling, provide an opportunity to address these barriers. Building on these tools, we developed a semi-automatic workflow that uses Thin-Plate Spline (TPS) transformation and magnetic resonance imaging (MRI) data to create personalized musculoskeletal models without requiring bone segmentation. Our workflow identifies biologically and

geometrically homologous points on template and target objects, enabling non-affine fitting of generic models to individual anatomy. Unlike Statistical Shape Modeling approaches [28,29], which match large numbers of vertices on surfaces between bone shapes and require substantial reference populations, our method identifies a smaller set of biologically and geometrically homologous landmarks on template and target anatomy [30,31,32]. This landmark-based approach enables non-affine 3D transformation [30,33] that maps both bone geometry and muscle-tendon paths from generic models onto individual anatomy, creating models that are personalized in both skeletal and muscular morphology while remaining accessible to researchers without extensive computational resources or reference databases.

To evaluate this workflow and challenge the assumption that personalization is unnecessary for healthy adults, we compare MRI-based musculoskeletal models with generic-scaled models in eight healthy adults. We examined the impact of personalization using a number of parameters, including muscle moment arms, joint kinematics, joint moments, and joint contact forces during walking.

## Results

Each of the eight participants (four males and four females, healthy, from 26 to 51 years of age) is represented by two models: a generic-scaled Rajagopal (2016) model [34] that had its linear dimensions scaled to match the participant's anthropometry and an MRI-based model that was obtained with the help of our pipeline (see Methods and Supplementary information).

### Comparing musculoskeletal geometry between generic-scaled and MRI-based models

**Linear dimensions and angles.** Size and shape of segments were different between MRI-based and generic-scaled models of the same individual. Generic-scaled models underestimated the length of the femur by up to 15% and overestimated the length of the tibia by about 5–7% (Fig 1a). As a result, the femur-to-tibia ratio differed between two model types. In the generic-scaled models it was $0.92 \pm 0.040$ and in MRI-based models it is $0.99 \pm 0.033$. Pelvic dimension differences between the two models of the same person also reached 15%. Dimensions such as the distance between ischial bones were the most affected due to higher values in females. On average, the distance between ischial bones in generic scaled models was $98.0 \pm 5.0$ mm and in MRI-based models was $11.0 \pm 8.0$ mm. Finally, errors in generic-scaled joint locations were particularly high in some individuals. For example, P02 exhibited an incorrect estimate of the knee center location 30–35 mm higher in the generic-scaled model than it was allowed by the MRI femur length (Fig 1b).

Due to isometric scaling, femur and tibia angles (i.e., femur torsion, femur neck to shaft, and tibia torsion angles) were the same for all generic-scaled models irrespective of the individual. MRI-based model accounted for subject-specific values which led to differences between models of the same individual (Fig 1c). These differences, together with incorrect joint locations in generic-scaled models, resulted in variations in muscle path positions between generic-scaled and MRI-based models of the same person (Fig 1d). A superposition of joint locations and muscle path points for generic-scaled and MRI-based models of the same individual is given in the Supplementary Information (Fig A in S1 Appendix).

**Muscle moment arms.** The differences in muscle point locations influenced muscle moment arms during a gait cycle (Fig A in S1 Appendix). For comparison, we subtracted the absolute values of generic-scaled moment arms from the MRI-based ones and averaged the right and left sides. Around the hip, moment arms of five muscles were most affected over an average gait cycle: *tensor fascia latae* (mean difference $10.0 \pm 7.8$ mm), *sartorius* (mean difference $8.0 \pm 7.0$ mm), *gluteus maximus* medial part (mean difference $6.0 \pm 7.0$ mm), *adductor magnus* distal part (mean difference $5.0 \pm 8.0$ mm), and *psoas* (mean difference $4.0 \pm 7.0$ mm). Around the knee, the most affected muscles were *sartorius* (mean difference $9.0 \pm 6.0$ mm), *vastus internus* (mean difference $-6.0 \pm 4.0$ mm), *vastus lateralis* (mean difference $-5.0 \pm 2.0$ mm), *rectus femoris* (mean difference $-5.0 \pm 3.0$ mm) and *gastrocnemius medialis* (mean difference $5.0 \pm 2.0$ mm). Positive values of the differences indicate larger muscle moment arms in the MRI-based models, whereas large standard deviations reflect individual variation as well as the variation across the walking cycle time points.

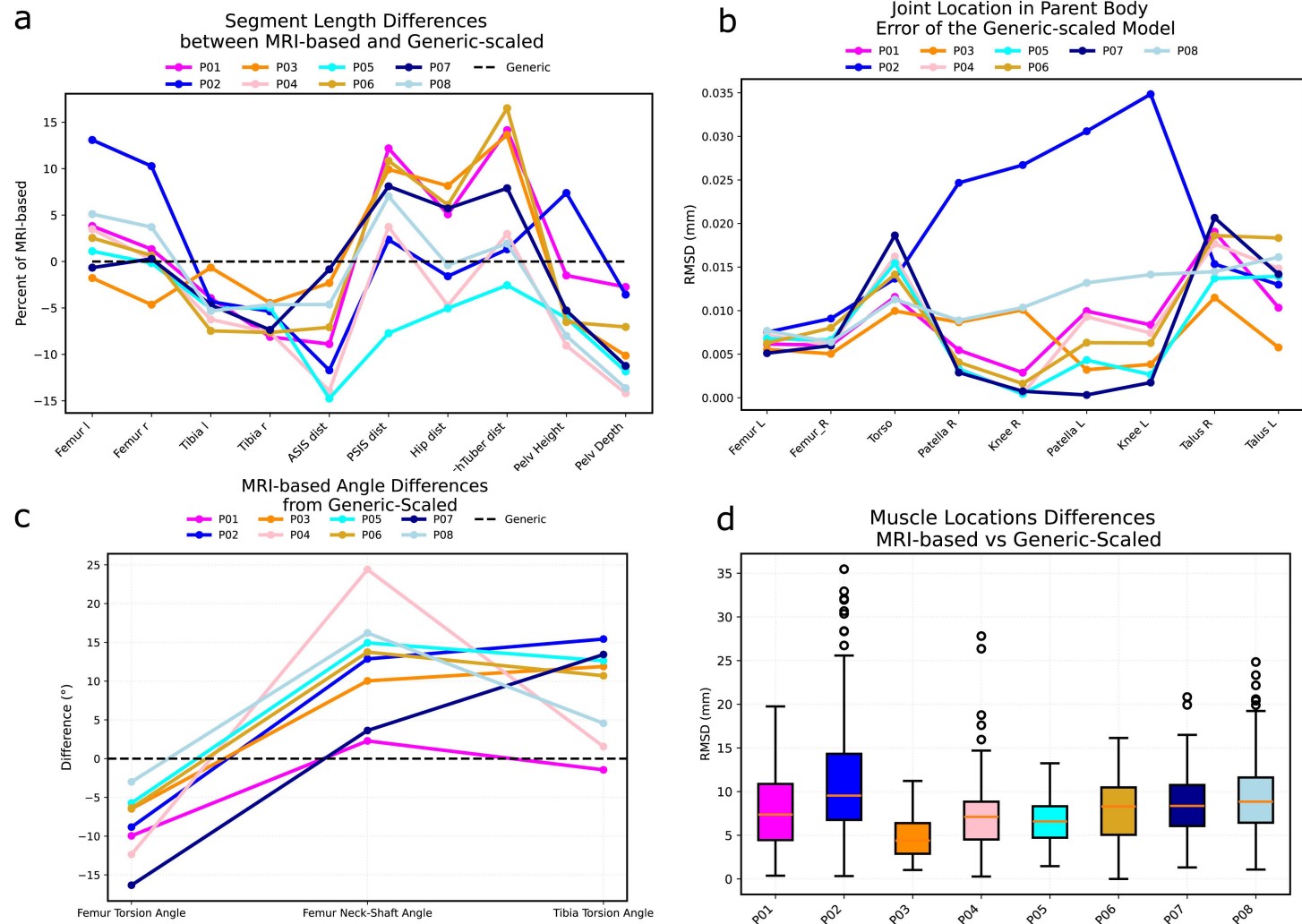

**Fig 1. Geometric comparison between MRI-based and generic-scaled models.** P01, P02, etc. labels indicate participants. **(a)** Differences in segment lengths in the MRI-based models from the generic-scaled ones, expressed as percentage of the MRI-based segment length; "ASIS dist": distance between anterior superior iliac spines on the MRI pelvis, "PSIS dist": distance between posterior superior iliac spines on the MRI pelvis, "Hip dist": distance between the centers of the hip joints, "IschTuber dist": distance between supero-medial points on ischial tuberosities, "Pelv Height": distance between the midpoint of the left and right superior ilia and the midpoint of the left and right inferior ischia, "Pelv Depth": distance between the superior pubic symphysis point and the torso origin point in the pelvis. **(b)** Differences in joint locations expressed as the Euclidean distance between MRI-based and generic-scaled coordinates of joint centers in the parent body. **(c)** Differences in femur and tibia bone angles between MRI-based and generic-scaled models, expressed in degrees. **(d)** Euclidean distances between locations of muscle origin, insertion, and via points for MRI-based and generic-scaled models for each individual. The differences are reported as absolute distances in mm.

**General trends in variation.** The first two principal components (PCs) described 63.9% of the variance (Fig 2a). PC1 (51.0% of variance) strongly differentiated between MRI-based and generic-scaled models with no overlap between the two groups. For the pelvis, this component differentiated between shapes with a narrow distance between the ischial bones and a relatively posterior position of the torso origin, as in the generic-scaled models, from those with a relatively wider distance between ischiopubic arches, wider distance between hip joint centers and a relatively anterior position of the torso origin, as in the MRI-based models (Fig 2b). For the limb segments, this component differentiated models with a relatively shorter femur and a longer tibia, as in the generic-scaled models, from those with a relatively longer femur and

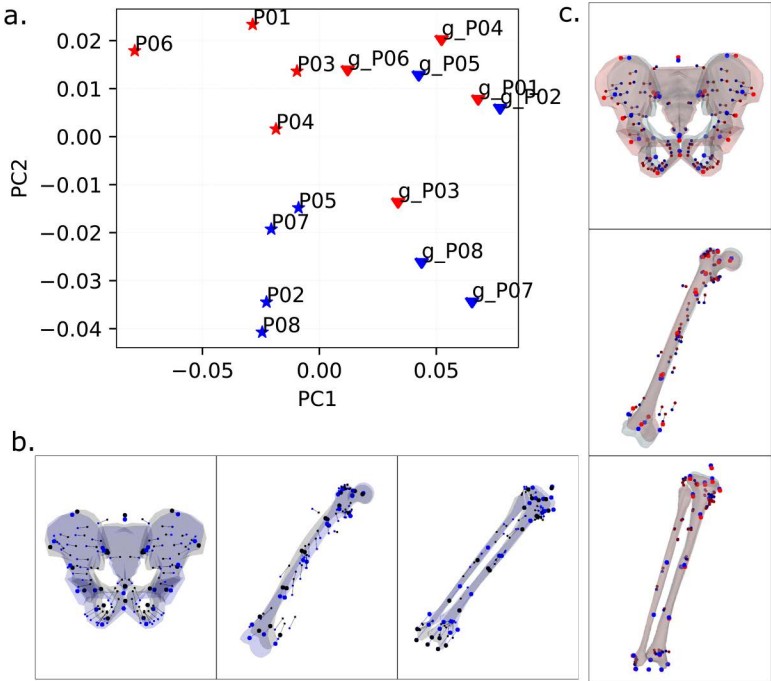

**Fig 2. (a) Principal component analysis of superimposed and standardized data for bone landmarks and muscle path points: the first two principal components; triangles indicate generic-scaled models, and stars show MRI-based models.** Male models are depicted in blue, and female models in red. **(b)** PC1 describes shape differences between generic-scaled and MRI-based models. Black points and a grey surface depict maximum positive values along PC1, representing generic-scaled models; blue points and a blue surface show minimum negative values along PC1, representing MRI-based models. **(c)** PC2 describes differences between males and females. Surfaces and points are plotted for the extreme positive and negative values, with red for females and blue for males.3D mesh visualizations were generated by ES using PyVista (open-source Python library, MIT License), PyVista license: https://github.com/pyvista/pyvista/blob/main/LICENSE.

a shorter tibia, as in the MRI-based models (Fig 2b). PC2 (12.0% of variance) differentiated MRI-based male and female models, where females tended to have a wider pelvis and a relatively shorted femur than males in our study cohort (Fig 2c). Generic-scaled models did not differentiate between males and females.

Principal component analysis highlighted that the differences between the two model types for the same individual were larger than the differences between individuals. This is evident from the percentage of variance (51.0%) described by the first principal component, as opposed to PC2 (12.0% of variance), which captured differences between individuals.

On further investigation, path point locations for some muscles around the knee in generic-scaled models were notably different compared to their locations in MRI-based models (Fig C in S1 Appendix). Muscle origin and via points of *gastrocnemius*, *adductor magnus*, *sartorius,* and *tensor fasciae latae* were located farther from the knee joint center in generic-scaled models compared to MRI-based models. The *sartorius* P2 via point was positioned much more anteriorly in the generic-scaled models than in the MRI-based models.

In many individuals, the patella in generic-scaled models was significantly larger than in MRI-based models, where the patella was scaled to match anteroposterior and mediolateral dimensions of the bone exactly. As a result, patellar via points for rectus femoris and vasti muscles differed by at least 1cm between the two model types in these individuals (Fig C in S1 Appendix).

### Comparing simulation results between generic-scaled and MRI-based models

Joint kinematics were similar between the generic-scaled and MRI-based models (Fig 3a), with mean differences of 0.80 ± 2.70° (mean ± SD). Net joint moments also showed small mean differences of -0.70 ± 2.20 Nm (Fig 3b). However,

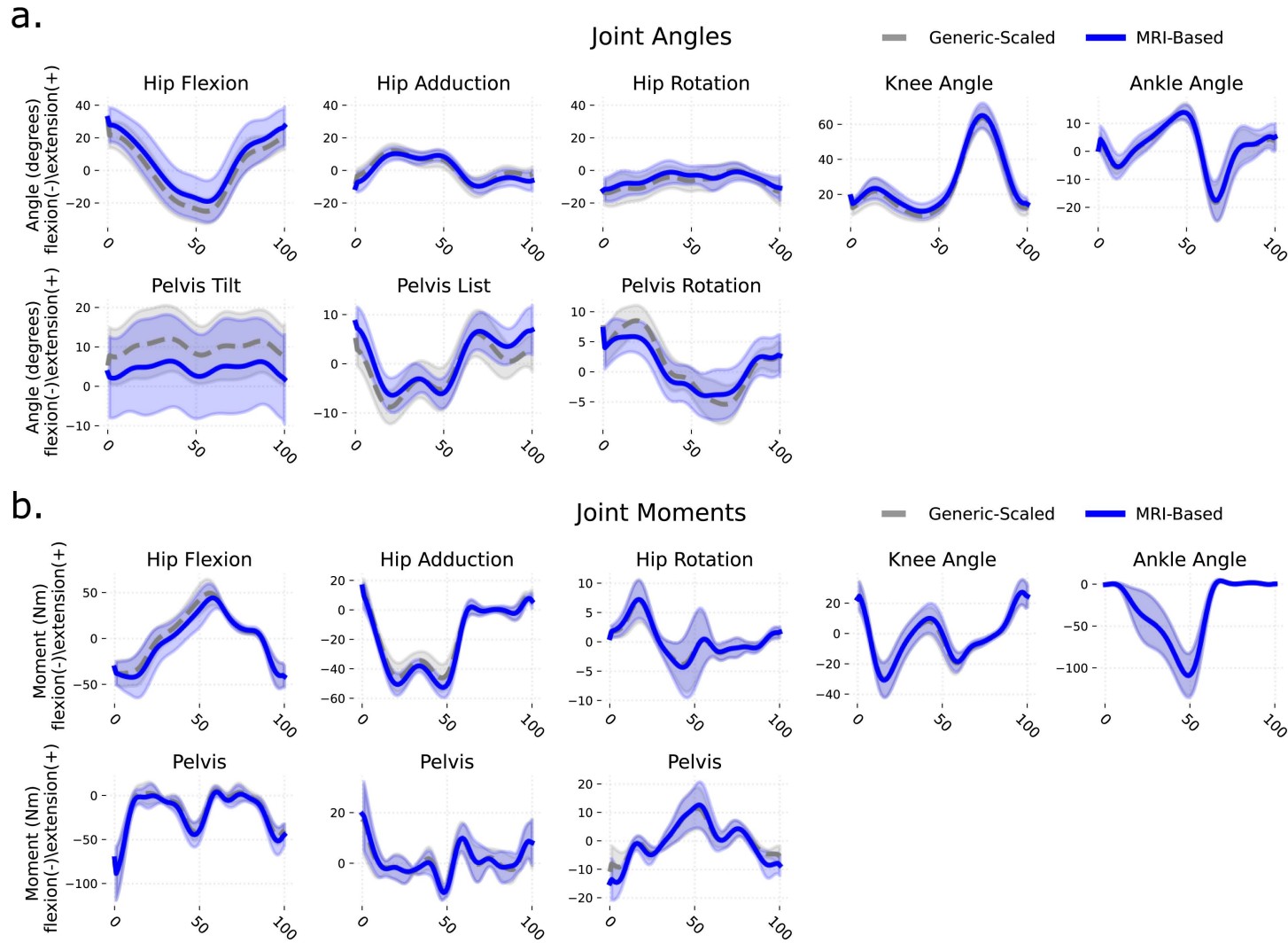

**Fig 3. Comparison of joint kinematics and joint moments waveforms of the right lower limb between MRI-based and generic-scaled models, averaged across all individuals. (a)** Joint Angles, **(b)** Joint Moments.

joint contact forces differed more substantially. During early stance, hip contact force (Hip Peak 1) was 0.73 ± 0.85 body weight (BW) higher in MRI-based models, while knee contact force (Knee Peak 1) was 0.84 ± 1.28 BW higher. Other peaks showed smaller differences between models (Fig 4b). Notably, the magnitude of these differences varied considerably across individuals (Fig D in S1 Appendix).

To investigate the general trend in the relationship between joint contact forces and bone morphology, we computed Pearson's correlations between peak joint contact forces and the first principal component (PC1), which differentiates between MRI-based and generic-scaled models based on pelvis, femora, patellae, and tibiae shape (Fig 5a). PC1 exhibited high negative correlations with first peak hip and knee joint contact forces. In other words, generic-scaled models produced lower joint contact forces than their MRI-based counterparts. The same pattern was observed when P06, the individual with the strongest expression of the geometric feature combination, as indicated by PC1 (Fig 1a), was excluded from the correlation test (Fig 5b).

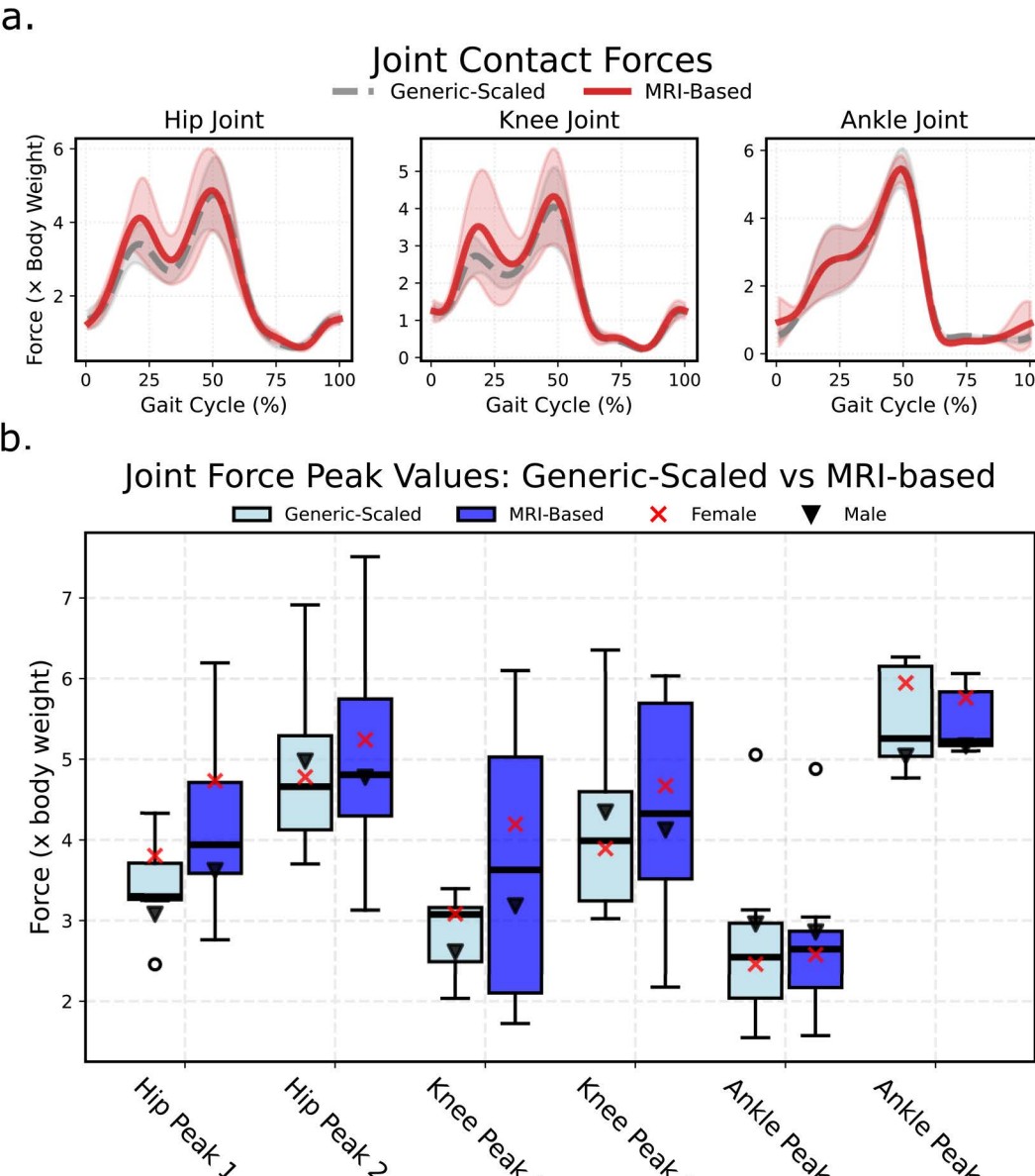

**Fig 4. Comparison of joint contact forces calculated with generic-scaled and MRI-based models. (a)** Mean waveforms with standard deviation; and **(b)** boxplot of values exported for first and second peaks of waveforms.

## Verification

Inverse kinematics yielded root mean squared (RMS) marker tracking errors of less than 20 mm for all individuals. Across all models, simulations yielded residual forces well below the recommended threshold [35] of 100 N (typically less than 10 N) and residual moments below 35 Nm (typically less than 5 Nm).

Despite discrepancies in leg length and muscle path point locations, muscle moment arms in the MRI-based and generic-scaled models showed similar consistentcy with with literature data in both profile and magnitude (Fig E in S1 Appendix).

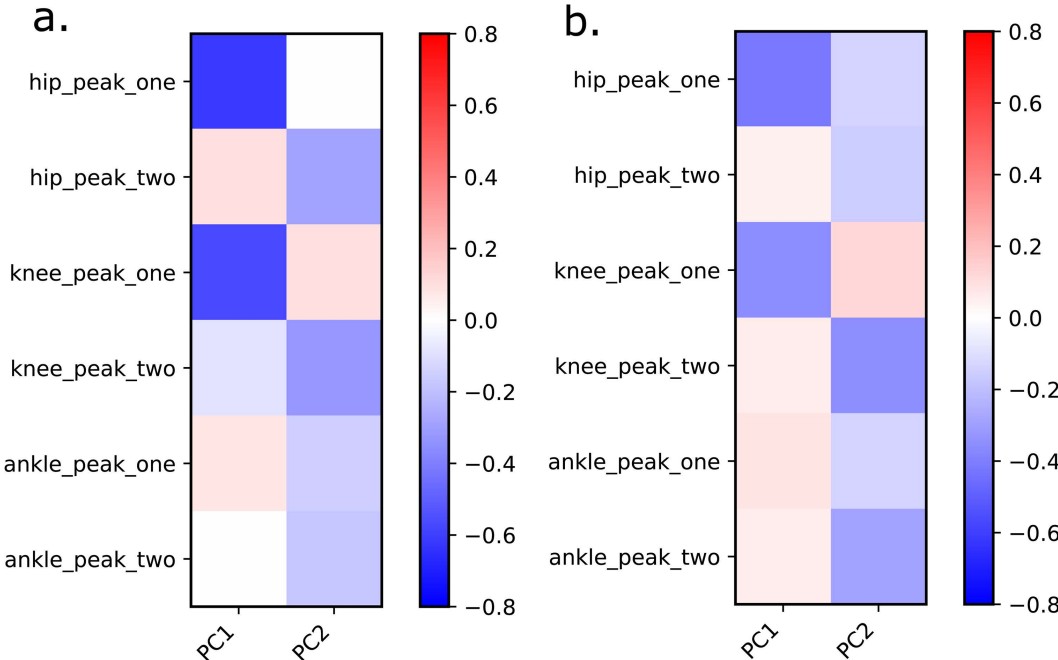

**Fig 5. Correlations between the shape of all models (MRI-based and generic-scaled) as described by PC1 peak joint forces. (a)** for the complete set of individuals, **(b)** excluding P06.

In most cases, EMG activity corresponded to muscle activations obtained from the generic-scaled and MRI-based models (Fig F in S1 Appendix), with activation patterns differing in magnitude but not in timing across the gait cycle. On average, the EMG signal registered earlier than estimated muscle activation, likely due to the electromechanical delay between muscle electrical activation, i.e., EMG, and mechanical output [36].

## Discussion

We developed a user-friendly, open-source pipeline for non-affine personalization of musculoskeletal models using 3D Slicer [26] and custom Python scripts. This workflow enables the creation of MRI-based models that closely match individual anatomy without requiring bone segmentation. Our results demonstrate that even in healthy adults, where generic models are often assumed to be sufficient, personalization reveals significant anatomical differences that impact biomechanical simulations.

The MRI-based models consistently differed from generic-scaled models in musculoskeletal geometry, with notable discrepancies in pelvis shape, femur length, shank length, and patella size. These discrepancies primarily arise from the proportions of the Rajagopal generic model, which has a relatively tall pelvis (tall iliac blades) compared to our sample of participants.

During generic scaling, femur length is typically scaled using the distance from the ASIS to the lateral femoral condyle—a distance that includes both pelvic height and femur length. Because pelvic height contributes a larger proportion of this distance in the generic model than in our participants, the scaling process underestimates true femur length in the generic-scaled models.

Patellar size was another notable discrepancy in geometry between generic-scaled and MRI-based models. The patella in the Rajagopal model is relatively large; consequently, when scaled linearly with femoral dimensions, it resulted in larger patellar dimensions in the generic-scaled models than in the MRI-based models for some individuals.

Consequently, most muscle paths in generic-scaled models deviated from corresponding muscle locations visible in MRIs. These differences influenced muscle moment arms, with *such muscles as tensor fascia latae*, *sartorius*, *gluteus maximus* medial part, *adductor magnus* distal part, and *psoas* (mean difference 4.0 ± 7.0 mm) with the largest discrepancies around the hip joint. Around the knee, *sartorius*, *vasti*, *rectus femoris*, and *gastrocnemius medialis* showed the largest deviations. Nevertheless, muscle moment arms in both generic-scaled and MRI-based models generally aligned with available empirical data. Importantly, MRI-based models produced valid kinematics and residual forces, sensu Hicks et al. [35]. Also, despite anatomical differences, kinematic outputs were broadly comparable between the two model types, except for pelvis tilt. This similarity likely stems from the use of identical anatomical coordinate systems and degrees of freedom in both models [37] as well as from our choice to maintain musculoskeletal parameters and segment inertial properties in the MRI-based models the same as in the generic-scaled variants. Consequently, joint moments showed strong agreement between model types, consistent with previous personalization studies [38].

Bosman et al. [39] showed that anatomical variability in muscle via point locations significantly affects calculated muscle forces in children with cerebral palsy, with gluteus medius, iliacus, and psoas being most sensitive. We observed similar sensitivity in healthy adults, though differences in gait patterns between populations may explain why other muscles (e.g., sartorius, gastrocnemius) were also substantially affected in our study.

Geometric differences between model types had pronounced effects on joint contact forces through two primary mechanisms. First, discrepancies in patella size and shape altered via point positions of the *rectus femoris* and *vastii* muscles, reducing their moment arms in MRI-based models. Second, several muscles (*gastrocnemius, adductor magnus, sartorius, tensor fasciae latae*) had shorter distances from their via points to the knee joint center in MRI-based models, similarly reducing moment arms. These reduced moment arms required higher muscle forces to generate equivalent joint moments, which contributed substantially to the elevated knee joint contact forces in MRI-based models.

MRI-based models exhibited greater variability in joint contact forces than generic-scaled models, with some individuals reaching 6.5 body weights—within typical modeling ranges [11,40,41] but exceeding instrumented implant measurements of 1–4 body weights [42–44]. The combination of pelvic, femoral, and tibial morphological features captured by PC1 was associated with elevated peak contact forces, consistent with previous findings that femur morphology (torsion angle, neck-shaft angle, femoral neck length) significantly influences joint loading [10,45].

Previous personalization studies in children and pathological populations have demonstrated improved physiological consistency with greater anatomical detail, showing that personalized geometry affects kinematics, joint moments, ligament lengths, and contact forcesl [14,38,46–50]. Notably, Kainz et al. [46] found geometry personalization had the greatest impact on simulation results in children with cerebral palsy using MRI-based models, while Modenese et al. [14] showed realistic knee contact forces could be estimated with properly scaled musculotendon parameters. Our study is the first to demonstrate that personalization also affects simulation outcomes in healthy adults, challenging the assumption that generic models are sufficient for this population. Unlike previous approaches using linear scaling or affine transformations, our non-affine method revealed systematic sex-specific differences in musculoskeletal geometry that influenced muscle activation patterns and joint contact forces.

Unlike previous studies that relied on linear scaling or affine transformations, our non-affine approach allowed us to reconstruct bone morphology and muscle paths with greater accuracy. This level of personalization revealed systematic differences between male and female musculoskeletal models, which may explain sex-specific differences in muscle activation patterns and joint contact forces. These findings underscore the importance of considering individual anatomy, even in healthy adults, to improve the accuracy and physiological relevance of musculoskeletal simulations.

Our study has several limitations. We did not account for individual motor control strategies; EMG-informed approaches may reduce excessive muscle activations [51]. Morphological changes in personalized models may require corresponding musculotendon parameter adjustments [52,53]. We did not modify segment inertial properties when creating MRI-based models, though we expect negligible impact. While we lack ground truth data for our participants, instrumented prosthesis

studies [11] suggest personalized models produce more accurate results. Despite these limitations, our findings demonstrate that personalization affects biomechanical outcomes even in healthy populations.

The model choice depends on the research question. Generic-scaled models may be sufficient for general kinematics-driven analyses, e.g., comparing joint contact forces between walking and running. However, imaging-based personalization becomes necessary when addressing anatomy-specific questions, e.g., how femoral anteversion differs between sexes and how this affects joint contact forces. Our paper highlights uncertainties that should be considered when using generic-scaled models.

## Conclusions

We presented a workflow for creating MRI-based personalized musculoskeletal models that closely fit individual skeletal and muscle geometries. Our findings demonstrate that MRI-based personalization substantially affects joint contact force estimations even in healthy adults. While generic-scaled models may be sufficient for general kinematic analyses, personalization should be considered when accuracy in joint loading estimates is critical.

Our workflow requires no bone segmentation or reference population and leverages open-source tools readily available to researchers and clinicians. We provide a step-by-step guide and Python code as supplementary material. Many clinical centers already have 3D motion capture and MRI data but lack tools to create personalized models. Our workflow bridges this gap, enabling more accurate biomechanical analyses with potential applications in treatment planning and clinical decision-making.

## Methods

### Ethics statement

Ethics approval was obtained from the Ethics Committee of the University of Vienna (reference number 00716). Written informed consent was obtained from all participants prior to data collection.

### Participants

Eight healthy individuals (four males and four females) volunteered to participate in this study. Participants' ages ranged from 26 to 51 (Table 1). Males had an average mass of 76.43 kg and body height of 1.84 m. Females had an average mass of 59.3 kg and height of 1.64 m.

Participants attended a 3D motion capture session at the Department for Biomechanics, Kinesiology and Computer Science in Sport at the University of Vienna (Vienna, Austria). This was followed by a magnetic resonance imaging (MRI) session at the Orthopaedic Hospital Speising (Vienna, Austria).

### Motion capture

Gait data were collected using a three-dimensional motion capture system (12-camera Vicon Motion Systems, Oxford, UK), five force plates (Kistler Instrumente AG, Winterthur, Switzerland), and a 16-channel electromyography (EMG) system (Cometa SRL, Bareggio, Italy). *Gluteus maximus*, *gluteus medius*, *tensor fasciae latae*, *biceps femoris*, *rectus femoris*, *vastus lateralis*, *vastus medialis,* and *gastrocnemius medialis* activation signals were collected on the left and right lower limbs using EMG sensors. An adapted Cleveland marker placement protocol [54] was used for tracking motion

**Table 1. Participants' statistics, mean (SD).**

|  | Number | Age [years] | Mass [kg] | Height [m] |
|---|---|---|---|---|
| **Males** | 4 | 26.75 (5.07) | 76.43 (10.32) | 1.84 (0.06) |
| **Females** | 4 | 36.25 (9.15) | 59.53 (5.49) | 1.64 (0.07) |

(Supplementary information, Table A in S1 Appendix). Participants walked across the lab stepping on three force plates at a self-selected speed. Seven trials with successful marker detection were selected for analysis.

## MRI images

Each participant underwent MRI scans of the pelvis and lower extremities (Siemens MAGNETOM Vida; Gradient Recalled Echo scanning sequence; segmented k-space variant; cardiac gating, respiratory gating, phase encode reordering, repetition time 5.42 msec; echo time 2.46 ms, magnetic field strength 3T; imaging frequency 123.25 MHz). Voxel size was $0.9 \times 0.9 \times 1$ mm.

## Model personalization

We used the Rajagopal et al. [34] model, a commonly used OpenSim model with arms removed and torso mass adjusted. For each participant, we created both a generic-scaled model (standard scaling from motion capture markers) and an MRI-based model (joint centers from MRI landmarks, muscle via points adjusted to individual geometry). Both followed ISB conventions. Bone surfaces of the MRI-based models were approximated for display purposes only. The complete pipeline with all scripts, detailed documentation, and a worked example is publicly available at https://github.com/katya-stanzy/thin-plate-spline_personalised_muscoloskeletal_model. Workflow steps 1–9 correspond to the pipeline (Table 2).

**Table 2. Reference for Methods sections and files in the repository\*.**

| Step | Notebook File | Description | Methods Section |
|---|---|---|---|
| 1. Extract static trial | example/model_update/1_extract_static_C3D.ipynb | Extracting a static trial from a C3D file. | Generic Scaled Models |
| 2. Scale generic model | example/model_update/3_scale_generic_model.ipyn b | Scale generic model to the skin markers using OpenSim scaling tool. Adjust muscle maximum force capacity. | Generic Scaled Models |
| 3. Create landmark template | example/mri/results/create_template.ipynb | Exporting a template of landmarks from a generic model. These have been originally placed on the model in OpenSim GUI. | MRI-based models |
| 4. Place landmarks on MRI | example/mri/results/orientation_with_tps.ipynb | Interactive placement of the template landmarks in the space of the participant's MRI with the help of 3D Slicer and a Jupyter plugin for it. | MRI-based models |
| 5. Initial muscle warping | example/model_update/2.0_use_mri_data.ipynb | Using MRI bone landmarks, warp generic model muscles and skin markers onto the bone data. Separate into segments and bring segments to ISB orientation. | MRI-based models |
| 6. Verify muscle paths | example/mri/control/control.ipynb | Interactive control of the muscle paths on the participant's MRI with the help of 3D Slicer and a Jupyter plugin for it. | MRI-based models |
| 7. Re-warp with corrections | example/model_update/2.1_after_controlling_mri_data.ipynb | Using MRI bone landmarks, warp generic model muscles and skin markers onto the bone data. Import new muscle locations from the Muscle Control step. Separate into segments and bring segments to ISB orientation. | MRI-based models |
| 8. Update model geometry | model_update/4_update_generic_model_with_mri_data.ipynb | Update the generic scaled model with the joint centers, references to bone surfaces and muscle paths obtained in the previous step. Re-fit skin markers using OpenSim scaling tool. Check for muscle points overlap with wrapping surfaces and reduce wrapping surface radii if needed. | MRI-based models |
| 9. Optimize muscle paths | model_update/5_optimize_muscles.ipynb | Plot muscle moment arms over the whole range of motion for the generic-scaled and the new MRI-based model. Optimise muscle path point location if necessary. | MRI-based models |

\* All file paths are relative to the repository root. A complete worked example with sample data is available in the worked_example/ directory. Installation instructions and software dependencies are detailed in the repository README. Key supporting scripts and utilities are listed in Table D in S1 Appendix.

Flowchart in Fig B in S1 Appendix and the repository flowchart.png. Complete instructions, environment files, and example data are provided in the repository to facilitate reproduction. An experienced operator requires approximately 2 hours to create an MRI-based model using this pipeline.

**Generic-scaled models.** The generic Rajagopal model was scaled to match participant dimensions determined by reflective markers during static motion capture (Table B in S1 Appendix). Scaling was performed using the OpenSim scaling tool, which linearly updates x, y, and z segment dimensions and scales muscle fibres' optimal length to individual segment dimensions. We also adjusted maximum isometric muscle force using a regression equation from Handsfield et al. [55] for calculating lower limb muscle volumes. The maximum muscle force scaling factor was determined by equation (1).

$$scaling\ factor\ =\ \frac{V_{scaled}\ /\ V_{generic}}{l_{optimal\ scaled}/l_{optimal\ generic}}$$

(1)

Where $V_{scaled}$, $V_{generic}$, are volumes of the generic and scaled models, $l_{optimal\ scaled}$ and $l_{optimal\ generic}$ are optimal fibre lengths of a specific muscle in the scaled individual and generic models. In their turn, the volumes were defined by equation (2) [54].

$$V\ =\ 47 \times mass \times body\ height\ +\ 1285$$

(2)

This equation agrees with the convention that maximum isometric muscle force is proportional to physiological cross-sectional area (PCSA). The scaled-generic model, together with muscle parameters and inertia values, became the template for creating the MRI-based models for each participant.

**MRI-based models.** A workflow to create the MRI-based model was developed using 3D Slicer software [26] and OpenSim [27] API in Python (Fig B in S1 Appendix). Creating the MRI-based model included updating the generic-scaled model with muscle paths and bone shapes obtained from MRI images. To obtain these data, we identified homologous landmarks on the generic model and participants' MRIs (folder/example/mri/results/create_template.ipynb in repository). These landmarks were chosen to represent each segment's geometry while providing sufficient guidance for later muscle path fitting. 106 landmarks were placed on the pelvis, femur, patella, tibia and fibula of the template model (Fig 5, Table C in S1 Appendix). Four further landmarks doubled the location of the hip and knee joints later in the pipeline for further expression in the child and parent reference frame of the segment, increasing the total number of landmarks to 110.

Landmarks were arranged into two groups. The primary group included 21 landmarks easily identifiable on MRI and essential for further steps, placed manually on MRI using 3D Slicer (Fig 6). Remaining landmarks were projected into MRI space using a Thin-Plate Spline (TPS) non-affine registration algorithm available in 3D Slicer (folder/example/mri/results/orientation_with_tps.ipynb). An operator verified the correct positioning of registered landmarks to ensure alignment with predefined orientation planes within the MRI volume. Once proper positioning was established, bone landmarks were used for Thin-Plate Spline fitting of muscle point paths and translations of muscle wrapping surfaces.

Fitting was achieved using a custom Python script (folder/example/mri/control/control.ipynb), though the same procedure can be set up using only 3D Slicer tools. Thin-Plate Spline bending strength was penalized at different levels for bone and muscle points (0.001), and bone surface (0.02). The penalty coefficients were determined experimentally to ensure that projected points, on the one hand, fell within the expected boundaries of bone and muscle positions and, on the other hand, the reconstructed surfaces were smooth for the display. Once projected, muscle path locations were rechecked to fit in the middle of the muscles' axial cross-sections in MRI space. Only a few manual adjustments were necessary. These changes were needed due to muscle origin and insertion points sometimes ending inside the bone and via path points falling outside of the muscle body center. The full list of the muscle adjustments is given in the Python notebook 'folder/example/mri/control/control.ipynb' in the project repository. The outcome of the step should contain all bone landmarks and muscle points in the 3D slicer reference frame (Fig 7).

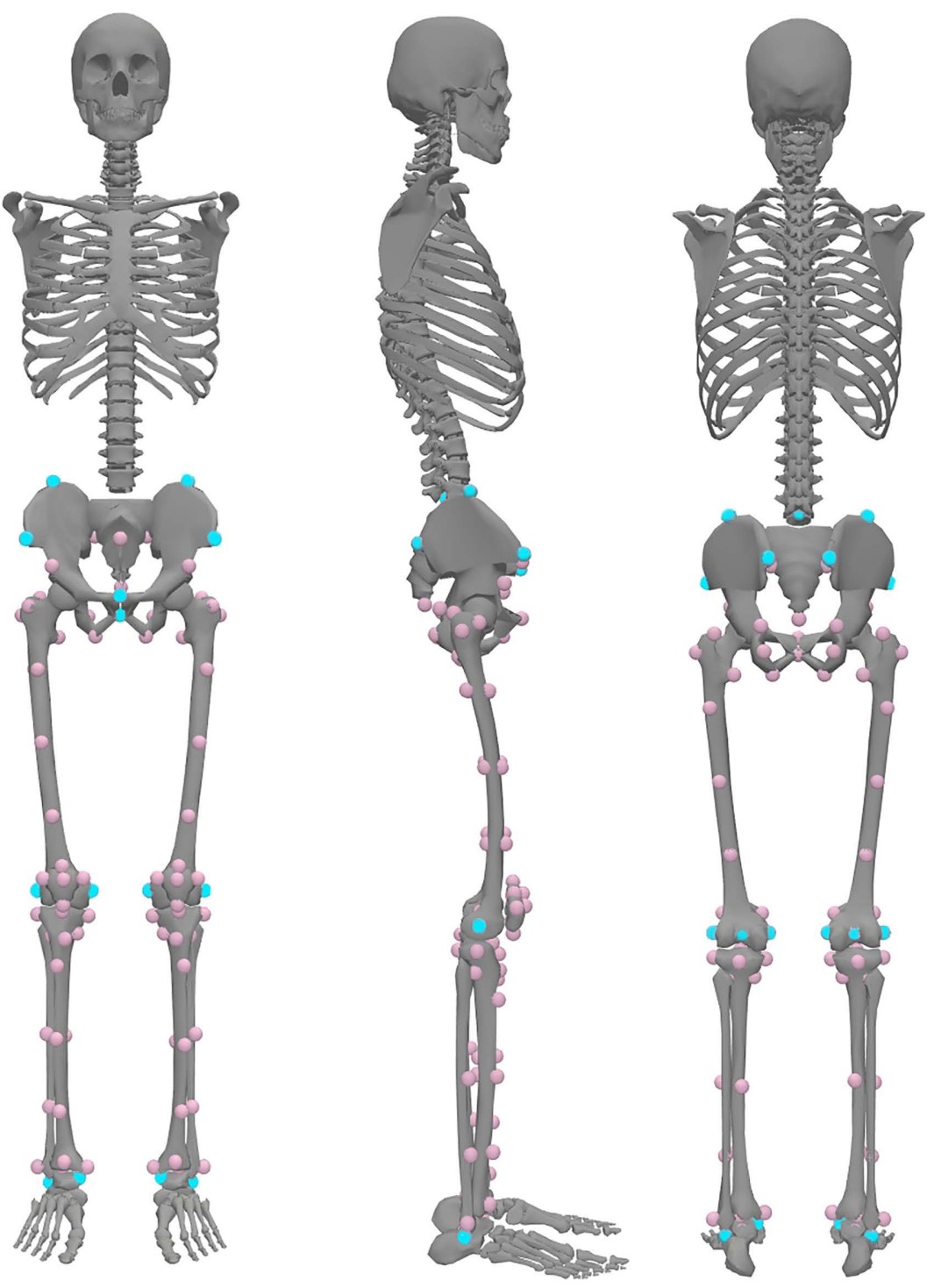

**Fig 6. Locations of homologous landmarks on the generic model.** The primary set of 21 landmarks is shown in blue. Landmarks in femur head centres are not visible. The secondary set comprises 85 landmarks shown in pink. Additional four landmarks were created in the center of the hip and the center of the knee on each side to allow for the joint localisation in the parent and the child segment body frames. The image was created by ES with the help of OpenSim [27], open-source musculoskeletal modeling software, Apache License 2.0, and Rajagopal et. al. (2016) model [34] (available at https://simtk.org/projects/full_body).

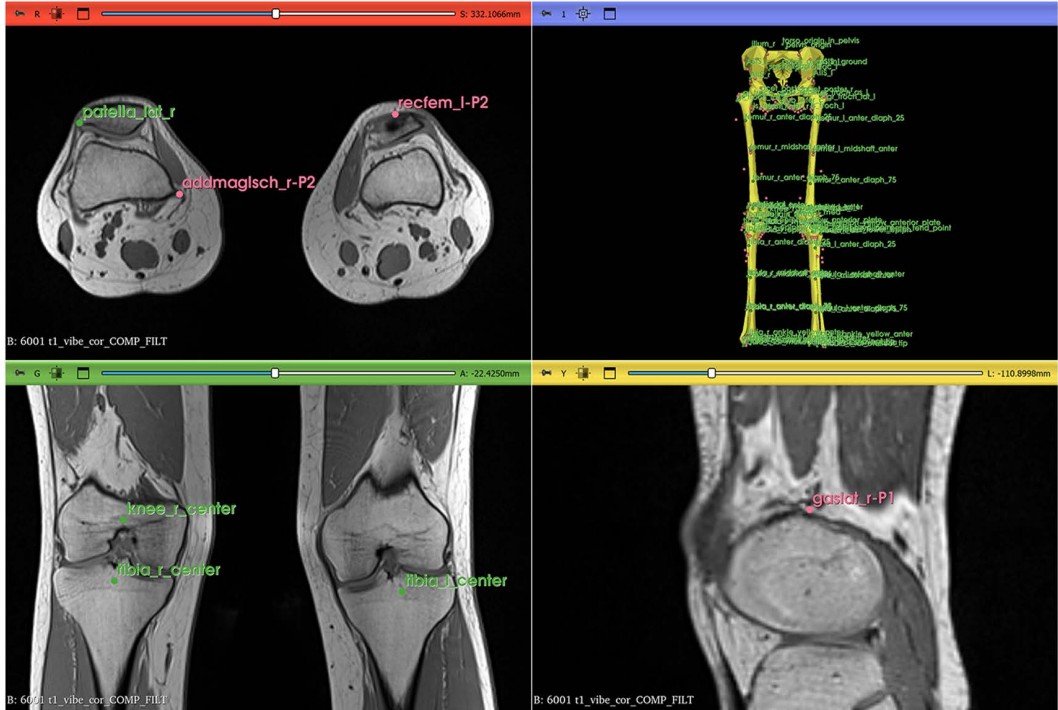

**Fig 7. Landmarks in 3D Slicer before data transformation into OpenSim coordinate sets.** Blue points show bony landmarks. Pink points show muscle paths. Yellow bone representations were created by warping the template OpenSim geometry onto the participant's landmark data. The image was created by ES with the help of 3D Slicer [26] (available at https://www.slicer.org/).

Next, the landmark data, i.e., 3D coordinates of joint locations, muscle paths and skin markers, were separated into segments (folder/example/model_update/2.0_use_mri_data.ipynb). Each segment was rotated to ISB-recommended orientation and scaled to meters to match OpenSim scaling convention. Simultaneously, new bone surfaces were created by warping template model bone surfaces onto participant data. Surfaces were rotated, translated into the ISB-recommended orientation, and later used for model display.

Once muscle paths, joint locations, muscle wrapping surface translations and bone surfaces were obtained and transformed to ISB standard orientation, a copy of the generic-scaled model was updated with this information (folder/example/model_update/4_update_generic_model_with_mri_data.ipynb). Wrapping surface radii, length, and orientation were preserved from the generic-scaled model. During adjustment, a custom Python script, a part of the "4_update_generic_model_with_mri_data.ipynb" notebook, verified potential intersections between muscle path points and wrapping surfaces to ensure accurate alignment. The wrapping surface radius was modified if the distance of the neighbouring via point from the axis of the wrapping surface was smaller than its radius. Skin marker locations underwent TPS transformation to fit the personalized geometry, after which experimental marker locations from static trials were fitted using the OpenSim scaling tool.

The skin markers were initially warped onto the MRI-based model using the same TPS transformation as was used for the muscle paths. These were then utilised as a template for the Inverse Kinematics fitting of the experimental markers with the help of the OpenSim scaling tool, where the scaling option was set to 'False' and marker mapping to 'True'. No further adjustments were made to the skin markers.

Personalization frequently results in discrepancies between muscle paths and wrapping surface size and position throughout the entire range of motion [56]. To address this, we created an additional optimisation script that adjusted muscle via point positions to prevent muscles from dynamically penetrating wrapping surfaces and avoid significant decreases in personalized muscle moment arms (folder/example/model_update/5_optimize_muscles.ipynb). The following objective function was minimised:

$$Optimized\ Value\ =\ d_{mean}\ +\ d_{max} + d_{shape} + d_{shift} + p_{drop}$$

where $d_{mean}$ means the average absolute distance between the generic and personalized muscle moment arm; $d_{max}$ means the maximum absolute distance between the moment arms; $d_{shape}$ is the term that calculates the absolute maximum height between the start and the end of the moment arm waveform; and $p_{drop}$ is the penalty for the sudden drop in the moment arm due to the interaction with the wrapping surface. We utilised an OpenSim API function calculating moment arm as the differential of the ratio between musculotendon length and joint angle [57]. Adjustment boundaries were customised for different muscles. For example, *gluteus maximus* via points could move within a 2 cm radius, while motion of *rectus femoris* and *vasti* points at the patella were limited to a 0.5 cm radius. The radius for each search space was determined by the muscle cross-sectional area in the vicinity of the point. For muscles represented by thin tendons, such as the patellar tendon, the search was limited to a smaller radius than for muscles with large cross-sectional areas like *gluteus maximus*.

We chose to keep musculotendon parameters (tendon slack length and optimal fiber length) and segment inertial properties in the MRI-based models the same as in the generic-scaled models to ensure that any observed differences between the two model types were due to geometry only. Initially, we did not expect to obtain noticeable differences in limb segment lengths between the two model types.

Assessing the reliability of the whole workflow was beyond the scope of this study and should be done in future investigations. However, we qualified the intra-observer reliability for locating bony landmarks. The mean difference across all bone landmarks for all individuals was 2.06 ± 1.86 mm. The mean difference across patella landmarks was 1.06 ± 0.37 mm.

## Analyses and verification

**Comparing generic-scaled and MRI-based models.** Musculoskeletal geometry of the generic-scaled and MRI-based models was compared by means of plotting differences in the segment lengths, joint locations, femur and tibia angles and in muscle locations (Fig 1). Femur and tibia segment lengths were computed as distances between adjacent joint centers. Pelvis dimensions included six measurements: distances between anterior superior iliac spines (ASIS), between posterior superior iliac spines (PSIS), between hip joint centers, between ischial tuberosities, pelvis height (superior ilium to inferior ischium), and pelvis depth (superior pubic symphysis to torso origin).

The shape of the femur and tibia was also assessed through three angles. Femur torsion angle was calculated as the angle between the femoral neck vector and the femoral condyle vector, both projected onto the plane perpendicular to the femur axis between hip and knee joint centres (biomechanical axis). Femur neck-shaft angle was calculated as the angle between the femoral neck vector and the proximal femur shaft vector (from 25% femur length to midshaft), both projected onto the frontal plane defined by the femoral head and medial/lateral condyles. Tibia torsion angle was calculated as the angle between the tibial condyle vector and the malleoli vector, both projected onto the plane perpendicular to the tibial biomechanical axis.

We calculated differences in muscle moment arms between MRI-based and generic-scaled models and report data for the five muscles with the largest differences around the hip and knee, averaged across left and right sides. Direct superimposition of joint locations and muscle paths from both model types in the ground frame are shown for each individual in Fig A in S1 Appendix along with representative examples of the resulting moment arm differences.

We compared overall trends in the shape variation by analyzing bone landmarks and muscle paths from generic-scaled and MRI-based models using principal component analysis (PCA). The combination of bone and muscle path points in one analysis is justified by their intrinsic anatomical connection, which was maintained through the non-affine TPS transformation. For this analysis, we extracted pelvis, femur, and tibia point locations in their local segment coordinate systems and normalized the combined point cloud to unit scale. Principal components were extracted from these superimposed and standardized landmark clouds after removing variations due to translation, rotation, and size [58]. The resulting principal components thus summarize major variations in segment shape and proportions.

All biomechanical analyses were performed using custom Python scripts and the OpenSim 4.5 API. The computational pipeline included inverse kinematics, muscle moment arm calculation, inverse dynamics, static optimization (minimizing the sum of squared muscle activations with equal weighting), and joint reaction analysis. Joint kinematics and dynamics were compared between MRI-based and generic-scaled models by plotting mean waveforms for each model type and calculating mean differences and standard deviations across the walking cycle.

As a primary outcome measure, we quantified the sensitivity of joint contact forces to model personalization. For hip, knee, and ankle joints, we report mean waveforms and standard deviations, compare peak forces during early and late stance phases, and calculate mean differences between model types across the walking cycle.

**Verification.** We compared kinematic tracking errors and residual forces with recommended threshold values [35]. Furthermore, model verification was performed by comparing muscle activations estimated by static optimization with EMG data collected from 16 muscles. Additionally, we compared muscle moment arms from generic-scaled and MRI-based models with data from the literature.

## Supporting information

**S1 Appendix.**
(PDF)

## Acknowledgments

We would like to acknowledge Adam Kewley and Ajay Seth, the developers of OpenSim Creator [32], for valuable discussions during the development of this work.

## Author contributions

**Conceptualization:** Ekaterina Stansfield, Hans Kainz.

**Data curation:** Ekaterina Stansfield, Willi Koller.

**Formal analysis:** Ekaterina Stansfield, Basílio Gonçalves.

**Funding acquisition:** Ekaterina Stansfield, Hans Kainz.

**Methodology:** Ekaterina Stansfield, Willi Koller, Basílio Gonçalves.

**Project administration:** Hans Kainz.

**Software:** Ekaterina Stansfield.

**Supervision:** Hans Kainz.

**Validation:** Ekaterina Stansfield, Willi Koller, Basílio Gonçalves, Hans Kainz.

**Visualization:** Ekaterina Stansfield.

**Writing – original draft:** Ekaterina Stansfield, Willi Koller, Basílio Gonçalves, Hans Kainz.

**Writing – review & editing:** Ekaterina Stansfield, Willi Koller, Basílio Gonçalves, Hans Kainz.

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
