## [Decision Letter · Decision Letter 0]

17 Dec 2025

PCOMPBIOL-D-25-01950

Do We Need Medical Imaging-Informed Musculoskeletal Models for Simulations in Healthy Adults? A New Workflow Based on Magnetic Resonance Imaging Highlights the Importance of Personalized Geometry

PLOS Computational Biology

Dear Dr. Kainz,

Thank you for submitting your manuscript to PLOS Computational Biology. After careful consideration, we feel that it has merit but does not fully meet PLOS Computational Biology's publication criteria as it currently stands. Therefore, we invite you to submit a revised version of the manuscript that addresses the points raised during the review process.

Please address the detailed points by reviewers regarding readability and improving the ease of reproducibility. In particular two reviewers have made comments regarding the github which should be made more accessible  to align with the policies of this journal.

We look forward to receiving your revised manuscript.

Kind regards,

Emma Claire Robinson

Academic Editor

PLOS Computational Biology

Marc Birtwistle

Section Editor

PLOS Computational Biology

**Journal Requirements:**

Potential Copyright Issues:

i) Figures 1b, 5, S1, and S2. Please confirm whether you drew the images / clip-art within the figure panels by hand. If you did not draw the images, please provide (a) a link to the source of the images or icons and their license / terms of use; or (b) written permission from the copyright holder to publish the images or icons under our CC BY 4.0 license. Alternatively, you may replace the images with open source alternatives. See these open source resources you may use to replace images / clip-art:

5) We note that your Data Availability Statement is currently as follows: "All relevant data are within the manuscript and its Supporting Information files.". Please confirm at this time whether or not your submission contains all raw data required to replicate the results of your study. Authors must share the “minimal data set” for their submission. PLOS defines the minimal data set to consist of the data required to replicate all study findings reported in the article, as well as related metadata and methods (https://journals.plos.org/plosone/s/data-availability#loc-minimal-data-set-definition).

1) If the funders had no role in your study, please state: "The funders had no role in study design, data collection and analysis, decision to publish, or preparation of the manuscript."

7) Please revise your current Competing Interest statement to the standard "The authors have declared that no competing interests exist."

**Reviewers' comments:**

Reviewer's Responses to Questions

**Comments to the Authors:**

**Please note that two reviews are uploaded as attachments.**

Reviewer #1: The submitted manuscript describes and attempts to validate/verify the resulting models by comparing results from typical musculoskeletal modelling steps. I would like to start by congratulating the authors on the mammoth tasks of the work they have presented here – not only in the actual simulations but also in the documentation provided and sharing of the workflow. This is the kind of work needed to continue to elevate the field of biomechanics. I nevertheless have some comments and questions based on the submitted manuscript which I believe warrant some form of discussion or at minimum a clarification. I have listed my “major” comments in general comments sections and smaller grammar or inconsistencies in the “specific comments”. I look forward to receiving an updated manuscript and the responses to my questions. You can find my comments and questions in the attached document.

Reviewer #2: GENERAL COMMENT

This paper presents on a novel pipeline to enable the faster generation of personalized musculoskeletal models from medical imaging and motion capture data, exploiting various software and without requiring images to be segmented beforehand. The entire pipeline was tested on data from 8 healthy adults, and a working example is provided at the project page on GitHub (link available in text).

This topic is of particular interest to the biomechanics community, as demonstrated by the number of workflows and pipelines developed over the last few years to develop musculoskeletal models with different degrees of personalization. Indeed, generic scaled models are typically chosen to minimize the efforts, both in terms of cost and time required for data acquisition and processing, but fail to capture many features that can make a difference on the simulations. This is more apparent for subjects with musculoskeletal or neuromuscular disorders but holds true for healthy individuals.

The rationale for this study and for the development of a new pipeline to generate personalized musculoskeletal models is clear and well laid out, and it is understandable. The reported results are encouraging. However, the introductory section could be improved and expanded to include some more references to current/existing pipelines to develop musculoskeletal models, possibly highlighting their main limitations. Furthermore, the methods section lacks the level of detail required to ensure a full understanding and to replicate the procedure. Consequently, also the discussion section should be revised, as I believe that some key points are not sufficiently discussed.

Please see specific comments below.

SPECIFIC COMMENTS

- Abstract -

Please consider adding some numbers to quantify the observed differences, even in the abstract.

- Introduction -

The introduction is good in general but could be improved. One critical aspect that is not touched on is how could one justify the need for acquiring MRI data on healthy participants, considering the costs and the organizational issues that may be connected to it.

>Lines 97-100: Additionally, models’ muscle fiber parameters […] for routine use.

Comment: Please be more specific, for the readership to know. Consider adding some examples of data and parameters that can be extracted from medical imaging data. Also, it should be clearly said that some of these methods cannot be used for all muscles (e.g., ultrasound imaging for deep muscles).

Comment: To better stress this point (i.e., that methods for personalization are typically time-consuming and manual), provide some more details to the readers. For example, how much time is required roughly to collect and process medical imaging data, or for other pipelines to generate a musculoskeletal model?

>Lines 101-102: […] such as 3D Slicer for segmentation and annotation and OpenSim for musculoskeletal modeling […].

Comment: Please add references for both 3D Slicer and OpenSim, and to works using these tools to develop (personalized) musculoskeletal models – if any.

>Lines 109-111: […] but our approach as 3D Slicer for segmentation and annotation and OpenSim for musculoskeletal modeling […].

Comment: This paragraph does not add much to the introduction and repeats what came before. Consider removing it. Also, not many readers may be familiar with the Thin-Plate Spline transformation: please add some references. Please be careful in using the term ‘fully personalised’.

- Results -

>Lines 121-122: Each of the eight participants (four males and four females, healthy, from 26 to 51 years of age) […].

Comment: Provide more details on the tested population. The information in Table 1 can be easily included here in brackets.

>Line 128-129: Bony landmarks and muscle paths of generic-scaled and MRI-based models were analysed using principal components.

Comment: Pease provide the rationale for combining both anatomical and muscle points in one single PCA.

>Lines 140-143

Comment: Not all these features can be easily appreciated on the figure. Would it be possible to add some zoomed images to highlight lines between points or black dots?

>Lines 149-151: In most cases, […] (Fig. 1c).

Comment: Is this to be attributed to the generic geometries in themselves, to a misplacement of the markers on the generic unscaled model and/or on the subject in the gait lab, or to both? Please comment.

>Lines 156-160.

Comment: How critical is it that muscle via points on generic scaled models are located further away from the knee joint centre? Does it have to do with different underlying geometries and/or the absence of wrapping surfaces?

>Lines 163-168

Comment: How easily can one identify the landmarks on the patella? This could be a possible explanation for the large reported errors.

>Lines 172-173: Knee and subtalar angles exhibited the least difference between models, and pelvis tilt was affected by the personalisation the most.

Comment: Please be more specific and quantify such differences. It is normal for some pelvis angles to be ‘off’ in MRI-based models, considering the supine position of the subjects in the scanner. Although, it should be later corrected.

>Figure 2

Comment: Please report also the pelvis list, tilt and rotation angles.

>Lines 208-209

Comment: Is it any different for the generic-scaled models? Please specify.

>Line 212-213: Despite […] muscle moment arms […] similar to generic-scaled ones.

Comment: Similar how? In profile, values, both?

- Methods -

In general, the authors well described the pipeline and steps required for model personalization, although some additional details should be provided. For example, there is no mention of the time required to collect and process the experimental data nor to generate the models using the proposed pipeline. This is quite relevant. In addition, the authors do not explain if and how muscle parameters (i.e., tendon slack length and optimal fiber length) were updated in the MRI-based models, which is critical.

>Table 1

Comment: Please report individual subjects’ information in here. Mean and standard deviation can be directly added in text (as suggested in other comments).

>Line 327: Gait data were collected […].

Comment: Please be more specific. What kind of task were the subjects asked to perform (i.e., at which speed?) and how many trials were collected?

>Lines 335-338.

Comment: Please add the name of the scanner manufacturer.

>Line 358: = 47 × × ℎ ℎ + 1285

Comment: While the use of this formula makes perfect sense for the scaled model, I am not sure it provides a good approximation of the generic volumes on the unscaled model. I would have computed the generic model’s volumes by inverting the formula of the maximal isometric force, as all parameters are known. Please comment.

>Lines 360-362: The scaled-generic model, together with muscle parameters and inertia values, became the template for creating the MRI-based models for each participant.

Comment: It seems the generic inertial parameters were used as template for the personalised musculoskeletal model, which can be considered a limitation (especially if the generic scaled geometries differ from the real bony geometries). Please comment and provide a rationale for this choice. The inertial parameters likely affect the system’s dynamics.

>Lines 371-376: The primary group included 21 landmarks easily identifiable on MRI and essential for further steps, placed manually on MRI using 3DSlicer (Fig. 5). Remaining landmarks were projected into MRI space using a Thin-Plate Spline (TPS) non-linear registration algorithm available in 3DSlicer. An operator verified the correct positioning of registered landmarks to ensure alignment with predefined orientation planes within the MRI volume.

Comment: These steps appear to be very much operator-dependent in terms of final outcome. Have the authors performed an analysis to quantify this? Furthermore, I think it is quite important to provide the time expenditure to complete the manual identification of the 21 landmarks.

>Line 378

Comment: Please provide a rationale for the simultaneous use of via points and wrapping surfaces.

>Lines 382-384

Comment: Please ensure numbers are consistent. Earlier, in text, it was mentioned that 106 points were identified (now 21+89 = 110).

>Lines 389-390: Only a few manual adjustments were necessary (Fig. 6).

Comment: Please be more specific, as Figure 6 does not seem to show the manual adjustments.

>Line 403: During adjustment, a custom Python script […].

Comment: This script is not available online, but I think it should be added to the folder.

>Lines 421-423: Adjustment boundaries were customised for different muscles […].

Comment: Please explain why, and how the boundaries were defined.

>Lines 447-448: Furthermore, model validation was performed by comparing muscle activations estimated by static optimisation with EMG data collected from 16 muscles.

Comment: Could the authors please add some more details on the implementation of the static optimisation? Was it a weighted optimisation or the traditional equation?

- Discussion -

>Lines 231-236: […] sensu Hickes et al.

Comment: Please amend. It should read Hicks et al.

>Lines 231-236: […] except for pelvis tilt.

Comment: Please discuss more about this point, which is not unexpected, but that was largely not commented on nor quantified in the Results section.

>Lines 231-236: Joint moments also showed strong agreement, consistent with previous personalization studies

Comment: Please discuss on the impact of the inertial parameters that, from my understanding, were not updated in the MRI-based (vs generic scaled) models.

>Lines 248-249: Overall, MRI-based models exhibited greater variability in joint contact forces than generic scaled models, with estimates for some individuals reaching up to 6.5 body weights

Comment: Please report the mean and standard deviation when using MRI-based vs generic scaled models. It would be more informative and would better support the sentence.

>Lines 283-285.

Comment: I agree, but anatomy is not the only important aspect in musculoskeletal models. Especially when one is interested in predicting muscle forces and joint contact forces. In fact, these quantities are largely affected by how muscles work, which depends on the internal muscle parameters too.

Comment: Last, I believe that the Discussion section should report on the time expenditure and – if performed – on the operator dependency.

Reviewer #3: Review uploaded as attachment

**Have the authors made all data and (if applicable) computational code underlying the findings in their manuscript fully available?**

The PLOS Data policy requires authors to make all data and code underlying the findings described in their manuscript fully available without restriction, with rare exception (please refer to the Data Availability Statement in the manuscript PDF file). The data and code should be provided as part of the manuscript or its supporting information, or deposited to a public repository. For example, in addition to summary statistics, the data points behind means, medians and variance measures should be available. If there are restrictions on publicly sharing data or code —e.g. participant privacy or use of data from a third party—those must be specified.requires authors to make all data and code underlying the findings described in their manuscript fully available without restriction, with rare exception (please refer to the Data Availability Statement in the manuscript PDF file). The data and code should be provided as part of the manuscript or its supporting information, or deposited to a public repository. For example, in addition to summary statistics, the data points behind means, medians and variance measures should be available. If there are restrictions on publicly sharing data or code —e.g. participant privacy or use of data from a third party—those must be specified.

Reviewer #1: Yes

Reviewer #2: **No:** Some custom scripts mentioned in the manuscript do not appear to be in the online GitHub pageSome custom scripts mentioned in the manuscript do not appear to be in the online GitHub page

Reviewer #3: Yes

PLOS authors have the option to publish the peer review history of their article (what does this mean? ). If published, this will include your full peer review and any attached files.). If published, this will include your full peer review and any attached files.

**Do you want your identity to be public for this peer review?** For information about this choice, including consent withdrawal, please see our For information about this choice, including consent withdrawal, please see our Privacy Policy ..

Reviewer #1: No

Reviewer #2: No

Reviewer #3: No

**Figure resubmission:**
---

## [Decision Letter · Decision Letter 1]

26 Feb 2026

Dear Assoc. Prof. Dr. Kainz,

We are pleased to inform you that your manuscript 'Do We Need Medical Imaging-Informed Musculoskeletal Models for Simulations in Healthy Adults? A New Workflow Based on Magnetic Resonance Imaging Highlights the Importance of Personalized Geometry' has been provisionally accepted for publication in PLOS Computational Biology.

Best regards,

Emma Claire Robinson

Academic Editor

PLOS Computational Biology

Marc Birtwistle

Section Editor

PLOS Computational Biology

Reviewer's Responses to Questions

**Comments to the Authors:**

Reviewer #1: I would like to commend the authors on their attention to detail in address the concerns which I had with the initial submission. While, there are still remaining questions regarding the final conclusions made in this paper and the broader literature I believe they have done all within their power and all the can be considered reasonable to best verify and critically assess their presented workflow.

Reviewer #3: Thank you for taking the time to revise your manuscript. My previous comments were largely minor, and the authors have addressed them thoroughly.

**Have the authors made all data and (if applicable) computational code underlying the findings in their manuscript fully available?**

The PLOS Data policy requires authors to make all data and code underlying the findings described in their manuscript fully available without restriction, with rare exception (please refer to the Data Availability Statement in the manuscript PDF file). The data and code should be provided as part of the manuscript or its supporting information, or deposited to a public repository. For example, in addition to summary statistics, the data points behind means, medians and variance measures should be available. If there are restrictions on publicly sharing data or code —e.g. participant privacy or use of data from a third party—those must be specified.requires authors to make all data and code underlying the findings described in their manuscript fully available without restriction, with rare exception (please refer to the Data Availability Statement in the manuscript PDF file). The data and code should be provided as part of the manuscript or its supporting information, or deposited to a public repository. For example, in addition to summary statistics, the data points behind means, medians and variance measures should be available. If there are restrictions on publicly sharing data or code —e.g. participant privacy or use of data from a third party—those must be specified.

Reviewer #1: Yes

Reviewer #3: Yes

PLOS authors have the option to publish the peer review history of their article (what does this mean? ). If published, this will include your full peer review and any attached files.). If published, this will include your full peer review and any attached files.

**Do you want your identity to be public for this peer review?** For information about this choice, including consent withdrawal, please see our For information about this choice, including consent withdrawal, please see our Privacy Policy ..

Reviewer #1: No

Reviewer #3: **Yes:** Tyler CollingsTyler Collings

---

## [Editor Report · Acceptance letter]

PCOMPBIOL-D-25-01950R1

Do We Need Medical Imaging-Informed Musculoskeletal Models for Simulations in Healthy Adults? A New Workflow Based on Magnetic Resonance Imaging Highlights the Importance of Personalized Geometry

Dear Dr Kainz,

I am pleased to inform you that your manuscript has been formally accepted for publication in PLOS Computational Biology. Your manuscript is now with our production department and you will be notified of the publication date in due course.

With kind regards,

Judit Kozma
